# Acute on Chronic Thromboembolic Pulmonary Hypertension: Case Series and Review of Management

**DOI:** 10.3390/jcm11144224

**Published:** 2022-07-21

**Authors:** Isabelle Opitz, Miriam Patella, Olivia Lauk, Ilhan Inci, Dominique Bettex, Thomas Horisberger, Reto Schüpbach, Dagmar I. Keller, Thomas Frauenfelder, Nils Kucher, John Granton, Thomas Pfammatter, Marc de Perrot, Silvia Ulrich

**Affiliations:** 1Department of Thoracic Surgery, University Hospital Zurich, 8091 Zurich, Switzerland; miriam.patella@usz.ch (M.P.); olivia.lauk@usz.ch (O.L.); ilhan.inci@usz.ch (I.I.); 2Institute of Anaesthesiology, University Hospital Zurich, 8091 Zurich, Switzerland; dominique.bettex@usz.ch (D.B.); thomas.horisberger@usz.ch (T.H.); 3Institute for Intensive Care Medicine, University Hospital Zurich, 8091 Zurich, Switzerland; reto.schuepbach@usz.ch; 4Emergency Department, University Hospital Zurich, 8091 Zurich, Switzerland; dagmar.keller@usz.ch; 5Institute of Diagnostic and Interventional Radiology, University Hospital Zurich, University of Zurich, 8091 Zurich, Switzerland; thomas.frauenfelder@usz.ch (T.F.); thomas.pfammatter@usz.ch (T.P.); 6Clinic of Angiology, University Hospital Zurich, 8091 Zurich, Switzerland; nils.kucher@usz.ch; 7Division of Respirology, University Health Network, Toronto, ON M5G 2C4, Canada; john.granton@uhn.ca; 8Division of Thoracic Surgery, Department of Surgery, University of Toronto, Toronto General Hospital, Toronto, ON M5G 2C4, Canada; marc.deperrot@uhn.ca; 9Department of Pulmonology, University Hospital Zurich, 8091 Zurich, Switzerland; silvia.ulrich@usz.ch

**Keywords:** acute pulmonary embolism, chronic thromboembolic pulmonary hypertension, pulmonary endarterectomy

## Abstract

Chronic thromboembolic pulmonary hypertension (CTEPH) is a distinct form of precapillary pulmonary hypertension classified as group 4 by the World Symposium on Pulmonary Hypertension (WSPH) and should be excluded during an episode of acute pulmonary embolism (PE). Patients presenting to emergency departments with sudden onset of signs and symptoms of acute PE may already have a pre-existing CTEPH condition decompensated by the new PE episode. Identifying an underlying and undiagnosed CTEPH during acute PE, while challenging, is an important consideration as it will alter the patients’ acute and long-term management. Differential diagnosis and evaluation require an interdisciplinary expert team. Analysis of the clinical condition, the CT angiogram, and the hemodynamic situation are important considerations; patients with CTEPH usually have significantly higher sPAP at the time of index PE, which is unusual and unattainable in the context of acute PE and a naïve right ventricle. The imaging may reveal signs of chronic disease such as right ventricle hypertrophy bronchial collaterals and atypical morphology of the thrombus. There is no standard for the management of acute on chronic CTEPH. Herein, we provide a diagnostic and management algorithm informed by several case descriptions and a review of the literature.

## 1. Introduction

Pulmonary embolism (PE) is one of the most prevalent causes of death worldwide, accounting for 300,000 deaths per year in USA [1]. Clinical presentation varies and can include dyspnea, chest pain, pre-syncope or syncope, and hemoptysis, or can be discovered incidentally during diagnostic workup for another disease [2].

Chronic thromboembolic pulmonary hypertension (CTEPH) is one of the most serious of conditions that is encompassed in the “post-PE syndrome”, a term embracing the many sequela of PE [3]. CTEPH results from failure of resolution after deposition of thromboembolic material in the pulmonary vessels, transition of clots into permanent connective tissue, and a small vessel vasculopathy [4]. The diagnosis of CTEPH is based on hemodynamic and radiological criteria after at least 3 months of effective anticoagulation. Diagnostic parameters include a mean PAP of ≥25 mmHg along with a pulmonary arterial wedge pressure of ≤15 mmHg at right heart catheterization in patients with mismatched perfusion defects on V/Q lung scan. Specific diagnostic signs for CTEPH on CT angiography include ring-like stenoses, webs, slits, and chronic total occlusion [2]. This entity needs to be differentiated from patients with chronic thromboembolic pulmonary disease (CTED) if they present with persistent normal hemodynamics at rest and exercise limitations on the basis of chronic vascular occlusions. CTEPH and CTED account as entities within the spectrum of the post-PE syndrome. Computed tomography is essential in the diagnosis of CTED. Pre-existing CTEPH should be considered and excluded in all patients presenting with an “acute” PE. Patients presenting acutely to the emergency department with signs of PE may have true acute PE, sub-acute PE, or chronic PE [5]. Some patients present with signs of chronic disease at the time of PE diagnosis [6]. Up to 25% of patients with CTEPH have no known history of prior PE [7]. Usual clinical presentation of CTEPH is progressive exertional dyspnea leading to decreased exercise capacity; fatigue; or, in advanced stages, lower extremity edema and syncope as signs of right ventricle (RV) failure [8]. The onset of acute PE on preexisting CTEPH may be catastrophic, with rapidly worsening of RV function. Patients’ clinical condition may appear extremely severe, and a rapid evaluation and decision-making process is paramount. CTEPH/CTED diagnostic algorithm is shown in Figure 1.

Treatment of acute PE and CTEPH are mainly different: while the initial approach to PE is anticoagulation, rescue thrombolytic therapy, or surgical embolectomy in selected cases (high-risk patients, presence of septal defect) [2], CTEPH is amendable for surgical treatment with the removal of occluding material from pulmonary arterial circulation and possible restoration of normal pulmonary pressures. Surgical treatment is usually postponed to a non-acute, recompensated phase, but, in the context of severe clinical condition, differentiating acute PE from symptoms of pre-existing CTEPH may be difficult, and choosing the correct management might be challenging. A revised treatment algorithm may support the decision making, as shown in Figure 2.

We aimed to describe cases beyond the existing literature background in a series of patients who developed acute PE on CTEPH to elicit discussion on differential diagnosis, treatment, and timing of surgery for this severe and rare disease.

## 2. Clinical Cases and Results

### 2.1. Case 1

A 49-year-old man presented in the emergency department of a regional hospital with rapidly progressive dyspnea, orthopnea, and tachycardia. Echocardiography revealed a right heart decompensation with D-shaping of RV, a systolic pulmonary artery pressure (sPAP) of 50 mmHg, and a large thrombus in a persistent foramen ovale (PFO). The patient was referred to our institution after initiation of intravenous therapeutic anticoagulation. CT scan as well as a transthoracic echocardiography confirmed the thrombus in the PFO, showing a massive central, segmental, and subsegmental bilateral PE (Figure 3 and Figure 4), and a free-floating thrombus. Moreover, signs of underlying CTEPH were evident in CT images, such as the characteristic wall-adjacent, eccentric filling defects of the pulmonary vessels, enlargement of the pulmonary artery, and hypertrophy of the right ventricle with D-shape. Interdisciplinary indication for surgery was given for the treatment of the suspicion of “acute on chronic CTEPH” and for the removal of the thrombus in the PFO and contemporary closure of the PFO. Right earth catheterization was considered not as mandatory due to the severe clinical condition of the patient and the numerous evidence of disease at non-invasive tests. After cardiopulmonary bypass (CBP), pulmonary embolectomy with removal of acute and chronic embolic material and a formal pulmonary endarterectomy (PEA) were performed (Figure 5) in moderate hypothermia after closure of the PFO during the cooling phase. The patient could not be weaned from CBP, and therefore veno-arterial extra-corporeal membrane oxygenation (VA-ECMO) was initiated. Postoperatively, the patient recovered slowly after a moderate to severe systemic inflammatory response syndrome (SIRS) and was weaned from ECMO on postoperative day (POD) 6. After 15 days at the intensive care unit (ICU), the patient was transferred to the ward and discharged from hospital after 5 weeks. Long-term oral anticoagulation therapy was initiated (phenprocoumon and subsequent rivaroxaban). At their first visit, 2 months after surgery, he had NYHA I-II symptoms. The echocardiogram revealed hemodynamic improvement with an RV/RA (right ventricle over right atrium) pressure gradient of 24 mmHg and normal fractional area change (42%). Right heart catheterization after one year revealed no pulmonary hypertension with a mean pulmonary arterial pressure (mPAP) of 18 mmHg. At 6 min walking test, the patient performed 596 m with minimum oxygen saturation of 90%; peak VO2 at cardiopulmonary exercise test (CPET) was 42.2 mL/min/kg (72%).

### 2.2. Case 2

A 48-year-old man was admitted to the emergency department of another hospital for dyspnea and syncope. CT scan and echocardiography showed massive central, bilateral pulmonary emboli (Figure 6); severe RV dysfunction (Figure 7); and signs of CTEPH. He had an acute episode dyspnea, which was spontaneously resolved after a long flight a few months earlier. An in-hospital second event led to respiratory and heart failure despite prompt thrombolytic therapy. Because of ongoing shock, a VA-ECMO was initiated. Due to the necessity of reinsertion of the ECMO after initial successful weaning, the patient was referred to our hospital for surgical evaluation. On VA-ECMO, blood flow of 4.8 L/min, he had a mean mPAP of 34 mmHg. Medical therapy with sildenafil and ilomedin-inhalation did not lead to an improvement. Therefore, surgical treatment was discussed with our interdisciplinary team as a salvage procedure in case of missing treatment alternatives. Pulmonary endarterectomy and embolectomy were performed (Figure 8), and the patient was transferred back to the ICU on VA-ECMO. The post-operative course was complicated. Initially, there was a severe hemothorax treated with surgical evacuation. The resulting reperfusion edema had to be managed by upgrading to a V-VA ECMO. In addition, there was a persistent septic status with polymicrobial positive blood and bronchoalveolar lavage cultures. Further in the course, despite consistent therapeutic heparinisation, re-thrombosis of the left pulmonary artery occurred. Another salvage procedure, catheter-based local urokinase therapy, was initiated (Figure 9). Although this was not complicated by any bleeding, it failed to lyse the thromboembolic material. After a brief period of clinical stability, the infective status with viral and fungal infection became uncontrollable, leading to a septic shock. Together with the patient’s family, an interdisciplinary decision was made to proceed with palliative care given the uncontrolled infectious situation. The patient deceased 30 days post-operatively.

### 2.3. Case 3

A 55-year-old man with previous medical history of splenectomy due to trauma and previous episode of acute PE was presented to the emergency department of a peripheral hospital with worsening dyspnea and chest pain. The CT performed locally was sent to our hospital and discussed in our pulmonary embolism rescue team (PERT). CT showed central, segmental, and subsegmental PE with features compatible with acute PE and signs of pre-existing CTEPH. The initial evaluation reported a NYHA class III, with a sPAP of 100–110 mmHg, RV dilated with mild wall hypertrophy. After multi-disciplinary discussion and on the basis of the patient’s very good and stable general condition, as well as lack of symptoms and clinical signs of PH such as cyanosis or peripheral edema, the patient was placed on oral anticoagulation therapy as an outpatient treatment. Three months after acute admission, he presented clinically well compensated with no limitation in daily activities and a sPAP of 43 mmHg. Regular follow-up with a CTEPH expert was indicated to assess the need for surgical intervention. Right heart catheterization was performed after 6 months and one year on oral anticoagulation therapy. The latter showed a mPAP of 27 mmHg. CT scan confirmed nearly complete resolution of the bilateral emboli. Figure 10 shows CT scan sections before and after anticoagulation therapy. Recent CPET showed a VO_2_max at 21.3 mL/min/kg (corresponding to the 73% predicted), a work capacity of 112%, and a VE/VCO_2_ slope at 35.2. Currently, the patient is symptom free, which is why he has not been further evaluated for surgery. However, he will be followed carefully with regular visits at our PH clinic with CPET and echo evaluation.

### 2.4. Case 4

A 52-year-old woman with a known history of acute PE for 2 months, under oral anticoagulation treatment, presented to the emergency department with severe hemoptysis (300–500 mL). A bronchoscopy was performed, showing diffuse alveolar hemorrhage. Echocardiography showed signs of RV failure with sPAP of 84 mmHg and features of cor pulmonale (Figure 11). A CT scan revealed residual thrombus in the left pulmonary artery and radiological signs of CTEPH (Figure 9); moreover, a dilated pulmonary artery in comparison to the aorta was seen through transthoracic echocardiography (Figure 12). During hospital admission, a therapy with iloprost and sildenafil was implemented along with non-invasive ventilation. Despite treatments, clinical status did not improve, and the patient continued to present episodes of hemoptysis causing lung atelectasis and needing bronchial artery embolization. The case was discussed at the multidisciplinary CTEPH board, and surgical intervention was indicated, despite the decompensated status, on the basis of a diagnosis of acute cor pulmonale on CTEPH. Figure 13 shows pre-operative pulmonary angiography. A bilateral PEA was performed without intraoperative or postoperative complications (Figure 14). During the postoperative course, the patient did not experience any further episode of hemoptysis, and she was discharged on oral anticoagulation therapy after a careful stepwise evaluation performed during hospitalization. At one-year follow-up, the patient presented in good clinical condition, with no respiratory symptoms; echocardiography showed normal RV function, and right catheterization revealed an mPAP of 27 mmHg, PVR 3.4 WU.

Table 1 summarizes characteristics and clinical courses of the three patients treated with surgery.

## 3. Discussion

### 3.1. Acute on Chronic Disease

The concept of “acute on chronic” has been described by Ende-Verhaar and colleagues in the results of the InShape III study [11]. On the basis of the concept that a relevant proportion of patients with CTEPH and history of acute PE already had radiological signs of pre-existing CTEPH on the initial scan, they reviewed the imaging to score them, according to defined parameters, in order to identify signs of CTEPH at the time of PE episode. The authors were able to elaborate a score yielding a sensitivity of 70% and a specificity of 96% in predicting CTEPH diagnosis on initial CT scans. The most reliable explanation according to the authors was that CTEPH was already present at the time of the initial PE diagnosis with almost all cases having radiological signs of both acute PE and CTEPH.

In daily clinical practice in an emergency department, this is a condition that clinicians might face more often than expected. The treatment of choice of acute PE and CTEPH differs significantly: while anticoagulation therapy and in selected cases fibrinolysis or embolectomy are effective therapies for acute massive and submassive PE, in CTEPH, urgent PTE may be required to adequately unload the right ventricle in the case of failure of medical therapy [2,12]. Embolectomy alone does not improve prognosis in the case of CTEPH and may lead to major postoperative difficulties. Current guidelines [12,13] recommend PEA as the treatment of choice in patients with surgically accessible CTEPH that is potentially curative, with nearly normalized pulmonary hemodynamics and substantial clinical improvement seen in many patients [14]. In general, this is the critical element where an interprofessional team is required to aid in decision making for both the acute management and long-term follow-up and treatment.

Missing the differential diagnosis between the two conditions might delay correct referral and treatment [15,16,17] and, in case of acute and severe presentation, might cause failure to rescue of patients.

### 3.2. Misdiagnosis and Overlapping

CTEPH is a severe and potentially fatal disease with uncertain epidemiology. Inhomogeneity of the data regarding its incidence is related to the large proportion of underdiagnoses often leading to mistreatment. Usual clinical presentation is characterized by progressive onset of respiratory and systemic symptoms [17], which are mostly non-specific and often overlapping with other comorbidities [18,19]. Progressive exertional dyspnea, fatigue, syncope, and worsening peripheral edema may be present also in case of chronic pulmonary diseases, cardiac dysfunction, and obesity [15,18,19] and mask the CTEPH diagnosis.

A proportion of patients may have been misclassified because of a lack of history of symptomatic PE or deep venous thrombosis [3]. In fact, 19–63% of patients with confirmed diagnosis of CTEPH do not report a relevant past medical history [15,20] and might have suffered from silent PE or belong to the thrombogenic phenotype with in situ thrombosis [21].

However, in patients who can recall signs or symptoms of venous thrombosis or PE, and even in patients who suffered from confirmed episodes of acute PE, it is difficult to differentiate recurrent episodes of acute PE from true CTEPH in the absence of a specialist examination. The diagnosis of CTEPH is based on findings obtained after 3 months of effective anticoagulation in order to discriminate this condition from “subacute” PE [13]. These findings are mPAP ≥ 20 mmHg with PAWP ≤ 15 mmHg and an elevated PVR, mismatched perfusion defects on lung scan, and specific signs at imaging studies. Surveillance programs after acute PE should be considered to screen for CTEPH in high-risk groups [9], but, to date, there is no clear guideline applicable in daily practice. As result, a quote of patients with episodes of PE and possible subsequent development of CTEPH remains under-investigated. To this, in an emergency department setting, we should add a proportion of patients referred with some signs or symptoms of PE that were underestimated and consequentially not treated. These patients present clear risk factors for CTEPH but have not enough data to confirm the diagnosis.

On the other hand, CTEPH may often be misclassified as pure acute PE [3]. In a well-designed study, Guerin and colleagues [22] retrospectively reviewed laboratory tests, CT scans, and echocardiography of 146 patients at the time of acute PE diagnosis: of the seven patients lately diagnosed with CTEPH during follow-up, they were able to identify several signs that CTEPH might have been already present at the time of acute PE. Mainly, the initial CT scans, re-interpreted by a blinded senior radiologist, showed at least two signs of CTEPH in all patients. Additionally, patients with CTEPH had significantly higher sPAP at the time of index PE (70 ± 20 mmHg), which is unusual in the context of acute PE [23,24].

Indeed, patients presenting to emergency departments with sudden onset of signs and symptoms of acute PE may already have a pre-existing CTEPH condition decompensated by the new PE episode.

### 3.3. Imaging

The workup for a patient with CTEPH should include a history, physical examination, echocardiography, electrocardiogram, pulmonary function testing, arterial blood gases, V/Q lung scanning, right-sided heart catheterization, and CT pulmonary angiography, and be outside of the acute setting exercise tests [14]. While CT pulmonary angiography (CTPA) is the investigation of choice for the diagnosis of acute PE, planar V/Q lung scan remains the main first-line imaging modality for CTEPH [13]. However, in expert hands, modern CTPA may be accurate for detection of CTEPH with 96.1% sensitivity, 95.2% specificity, and 95.6% accuracy [25,26,27].

Because of severe clinical conditions and the emergency department (ED) setting, we will focus on the two first-line assessment tools that are normally performed in these cases: CTPA and echocardiography.

### 3.4. Computed Tomography Pulmonary Angiography (CTPA)

The best technical conditions for CTPA delivery include short breath hold (3–5 s) acquisition, thin collimation and thin-slice (≤1 mm) reconstruction, and three-dimensional analysis [9]. It is the best modality for evaluation of the proximal extent of the thromboembolic material [28]. Analysis of maximum intensity projection (MIP) images improved the sub-segment vessel [3] with good performance in detecting lobar and segmental thromboembolic changes [3,28,29,30]. In comparison to digital subtraction angiography, CTPA provides additional information on mediastinum, lungs, and collateralization.

Some typical local vascular features of the chronic thromboembolic process have been identified, such as the complete obstruction of the vessels or partial filling defects, mostly eccentric and wall-adjacent. Calcifications, post-stenotic dilatation, intravascular bands, webs, and organized emboli also resemble CTEPH [31]. By contrast enhancement, central and eccentric filling defects surrounded by high-attenuation contrast medium (polo-mint sign, railway track sign) are typical radiological signs of acute pulmonary embolism (Figure 15).

Typical CT systemic vascular characteristics are enlargement of pulmonary vessels, bronchial arteries, and neighboring vessels; the presence of pericardial effusion; and, most commonly, the enlargement and hypertrophy of the RV.

Furthermore, typical lung parenchymal characteristics are mosaic pattern, pulmonary infarctions, and bronchiectasis (localization correlates with the position of affected vessels [31].

There are indirect cardiac features for acute and chronic embolic processes, which can be evaluated by CT. 

In an acute setting, obstruction to pulmonary arterial flow and the increase in pulmonary vascular resistance causes increase in the RV afterload with RV dilatation, bowing of the interventricular septum, inferior/superior vena cava and azygos vein dilatation, and reflux of contrast in the inferior vena cava and hepatic veins [32]. Volumetric assessment of RV/LV ratio is the most accurate, although clinically cumbersome, CT parameter for identifying patients with RV dysfunction [33]. 

During the chronic process, increased pulmonary pressures result in RV hypertrophy characterized by free wall thickness > 5 mm, and over time, RV function deteriorates, resulting in RV enlargement. The deformation of the LV, “D shape”, is a result of the pressure differential between left and RV chambers.

In some CTEPH patients, there is a mismatch between thromboembolic burden and signs of RV dilatation on CTPA, i.e., little central filling defects but very large right ventricles, which is usually not seen in patients with acute PE.

### 3.5. Transthoracic Echocardiography

In both acute PE and in CTEPH, transthoracic echocardiography plays a role. In both conditions, it is used to identify RV dysfunction and PH. An estimation of the pulmonary pressure can be made by Doppler evaluation of the tricuspid regurgitant flow [34]. In the case of PE, findings might be used to score patients for increased risk of adverse outcome [35,36,37,38,39,40].

Additional CTEPH signs depend on the stage of the disease and include enlargement of the RV, leftward displacement of the interventricular septum, and encroachment of the enlarged RV on the left ventricle [41]. Contrast echocardiography may demonstrate a PFO with R-L shunt, the result of high right atrial pressures opening the previously closed intra-atrial communication [41].

Guérin and colleagues found that patients with underline CTEPH have significant higher sPAP at the time of the diagnosed PE [22]. They postulated that a sPAP > 60 mmHg may represent a sign of adaptation of the RV to a chronic obstruction of the pulmonary vascular bed, as it has been shown that the upper limit of pulmonary arterial pressure that can be acutely maintained by a normal RV is approximately 40 mmHg [24].

The precise level of pressure that a naïve RV can generate is uncertain, but most probably 50–55 mmHg should raise concern. However, it needs to be considered that some patients with acute PE have other conditions deriving from cardiac or lung problems that may also contribute to a chronically loaded RV and therefore higher RVSP. A sPAP > 60 mmHg may be useful to differentiate pure episodes of acute PE from an “acute on chronic” event.

### 3.6. Treatment Options

According to guidelines [13], the recommended treatment for CTEPH is PEA with circumferential removal of thromboembolic material to the level of subsegmental branches. The indication for surgical treatment of CTEPH should be discussed within a multidisciplinary team including surgeons, pneumologists, and radiologist experts in this specific disease [42]. Patients’ operability is one of the most challenging issues in the decision-making process, and it is judged mainly on the basis of the degree of accessibility of the thromboembolic material and should be conducted in a CTEPH expert center as described in the ESC ERS guidelines [2]. The success of PEA is inversely proportional to the degree of microvascular disease present and its contribution to overall PVR. Surgical expertise is required to ensure correct removal of the obstruction in the segmental and subsegmental vessels [43,44]. Neither the degree of RV dysfunction nor the severity of PVR are exclusion criteria for PEA, but patients’ comorbidities and life expectancy should be considered [45].

Surgical intervention is usually planned in a non-acute, recompensated phase, and after comprehensive evaluation, but in the case of severe RV dysfunction not responding to usual bridging therapies, PEA might represent a rescue and salvage treatment. Brener and colleagues recently described a successful PEA performed in the emergency setting during a cardiogenic shock for an acute PE unmasking the underlying CTEPH [46], as in our case of patient 2, where PEA was performed after repeatedly unsuccessful weaning from VA-ECMO [47].

In cases of inoperable disease, alternative strategies might be taken into account. Medical therapy might be offered with acceptable results [42,48]. Pulmonary microvascular disease in CTEPH partially resembles the distal vasculopathy found in PAH, and this provided the bases for “off-label” use of drugs approved for PAH [13]. These drugs include agents targeting the prostacyclin, endothelin-1, and nitric oxide pathways [49]; different studies showed a certain degree of clinical status improvement in CTEPH patients [50,51,52]. Pulmonary vasodilators are also used as bridge to PEA, to reduce surgical risk, or to overcome waiting lists [3].

Balloon pulmonary angioplasty (BPA) is an acceptable alternative for patients not eligible for surgery due to comorbidities or because of distal localization of the lesions [3,53,54,55,56]. Stepwise procedures in multiple sessions are necessary to minimize risk of morbidity and mortality [56]. Although promising, BPA should be considered only in inoperable patients since there are currently no data to support its use in proximal lesions as an alternative to PEA. BPA also plays a role for patients who underwent endarterecomty with residual symptoms and signs of PH due to distal lesions.

Lung transplantation has been considered an alternative option in selected cases with distal vessels disease considered not surgically accessible [42]. However, the line between accessible and inaccessible lesions has been moved more distally over time, contributing to reducing the number of lung transplantations compared to PEA for CTEPH [57]. Moreover, PEA has the advantage of not presenting the typical complications related to transplantation, such as rejection and immunologic and infective complications [57].

### 3.7. Cases Discussion: Decision-Making Process

Going back to our initial cases, we can make a critical evaluation of some emblematic examples of the decision-making process in the case of acute on CTEPH.

Patient 1 did not refer any relevant past medical history at the time of emergency department access. In this case, there was no previous evidence of CTEPH, and the diagnosis was made on the basis of radiological and actual clinical status. Because of the concomitant finding of patent foramen ovale, the multidisciplinary team proposed immediate surgical intervention in order to perform the PEA and to correct the cardiac defect. In fact, the systemic rewarming period after the deep hypothermic circulatory arrest and the sternotomy approach can be conveniently used for cardiac procedures [45].

Patient 2 referred to possible previous episodes of untreated DVT and PE. This finding in the past medical history may raise suspicion of a chronic disease behind the acute PE episode, but it is not sufficient alone for the formulation of a CTEPH diagnosis. This case falls in that subgroup of patients in whom the possible risk factors for CTEPH have not been identified and consequentially not treated. Despite the use of “off-label” drugs, clinical conditions continued to deteriorate, and PEA was offered as a rescue procedure. The decision was made against re-embolectomy due to the rather marginal general condition and the excessive burden of surgery. The final decision was made in favor of lysis therapy, although this is also associated with an increased risk profile. However, the patient did not overcome the postoperative period; he was never weaned from ECMO support and eventually died after several complications. Possibly, the patient was referred at a too late a stage of disease and concomitant complication, and uncontrollable coagulation disorder, despite effective anticoagulation, made a recovery impossible.

Patient 3 presented risk factors for CTEPH but was not referred to a specialist team and not fully evaluated for CTEPH. This misdiagnosis was probably due to the lack of clear guidelines to identify patients at risk for CTEPH development after the PE episode. A new episode of PE decompensated an already compromised RV causing a severe dysfunction. Expert opinion from a multidisciplinary team was then sought, and conservative management was recommended, with close observation by the CTEPH team and further evaluation. The patients recovered partially from the PE and remained under follow-up in a referral center for PEA to evaluate indication and timing for surgical intervention.

Patient 4 was under treatment for acute PE with anticoagulation therapy, but the rapid evolution of the clinical condition, including high sPAP at the time of diagnosis, suggested a pre-existing CTEPH. 

A period of 2–12 weeks of onset of symptoms is defined as subacute pulmonary embolism. Within this period, cardiovascular collapse and pulmonary angiogram showing massive pulmonary embolism (>50% obstruction in the major pulmonary arteries) are still missing [58]. However, surgical intervention is usually planned in a non-acute, recompensated phase; there is no optimal time for a surgical intervention in the case of subacute embolism and this strongly depends on the patient’s general condition and current cardiovascular sequelae. Therefore, at this stage, drug therapy is the frontrunner. In exceptional cases, characterized by a rapid clinical deterioration, surgery is preferable, despite the risk of reorganization of the thrombus material and the fragility of the thrombus.

The usual timing for CTEPH evaluation after the first symptomatic PE presentation has not been reached yet; thus, she did not have all the recommended tests for an accurate diagnosis. However, despite all the efforts made to restore a compensated status and proceed with the usual pathway, she did not recover, and the surgical intervention was proposed as a life-saving procedure. Postoperative course in this patient was brilliant, and the recovery was complete at follow-up.

## 4. Conclusions

CTEPH is a severe disease with mostly insidious onset of symptoms. Differential diagnosis is quite challenging, and this causes delay in correct referral and possible misdiagnosis. Some patients may present acutely with sudden deterioration of cardiac and respiratory function due to further episodes of PE or rapid decompensation of a pre-existing chronic condition. In the emergency department, it is challenging to distinguish between a pure acute PE and an “acute on chronic” episode. Recognizing the underlying condition may change the decision-making process. All efforts nonetheless need to be applied to restore a balanced clinical status to try to perform all the necessary pre-operative investigations and postpone the PEA to elective conditions. On the other hand, a surgical intervention might be lifesaving in patients not responding to bridging therapies and unable to be weaned off ECMO. In compromised patients with prohibitive co-morbidities, alternative strategies such as BPA can be taken into account.

The key point of these difficult cases is the multidisciplinary discussion within a panel of experts to ensure the best management possible in the emergency department, with a PERT team involving emergency medicine specialists, pulmonologists, cardiologists, radiologists, thoracic surgeons, angiologists, and cardiac surgeons. 

## Figures and Tables

**Figure 1 jcm-11-04224-f001:**
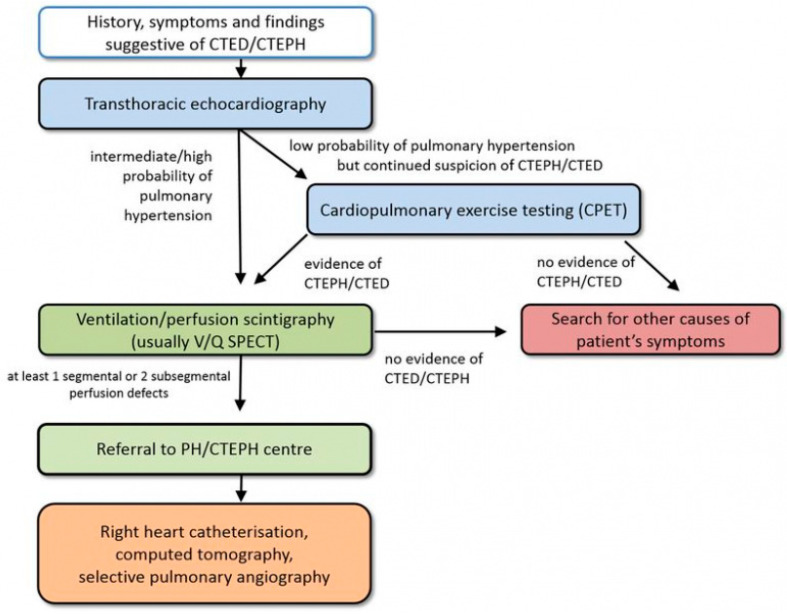
CTEPH diagnostic algorithm. Adapted from Wikens H, Ulrich S, consensus conference Cologne, 2016. CTED = chronic thromboembolic pulmonary vascular disease; CTEPH = chronic thromboembolic pulmonary hypertension; PH = pulmonary hypertension [9]. Reproduced with permission from Opitz, I.; Ulrich, S. Chronic thromboembolic pulmonary hypertension. Swiss. Med. Wkly. 2018, 148, w14702. https://doi.org/10.4414/smw.2018.14702.

**Figure 2 jcm-11-04224-f002:**
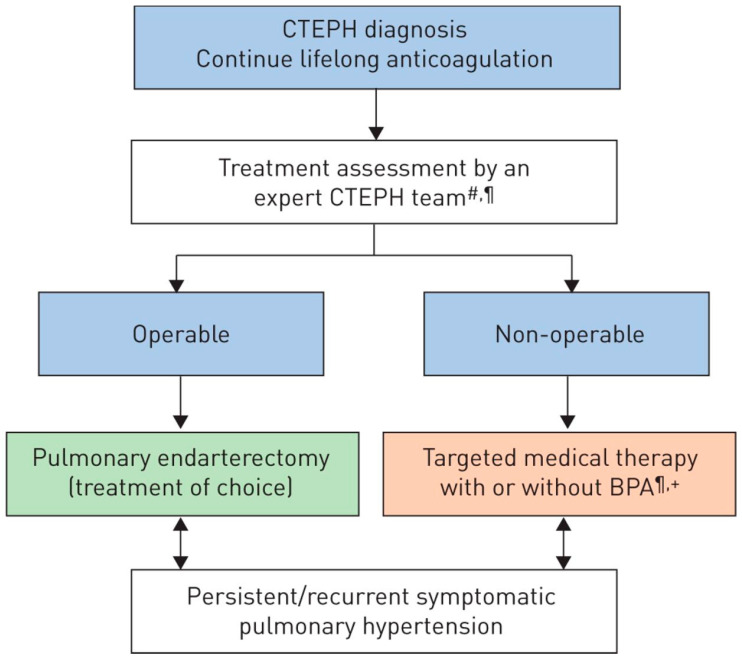
Chronic thromboembolic pulmonary hypertension (CTEPH): revised treatment algorithm. BPA: balloon pulmonary angioplasty. #: multidisciplinary: pulmonary endarterectomy surgeon, PH expert, BPA interventionist and radiologist; ¶: treatment assessment may differ depending on the level of expertise; +: BPA without medical therapy can be considered in selected cases [10]. Reproduced with permission of the © ERS 2022: European Respiratory Journal Jan 2019, 53 (1) 1801915. https://doi.org/10.1183/13993003.01915-2018.

**Figure 3 jcm-11-04224-f003:**
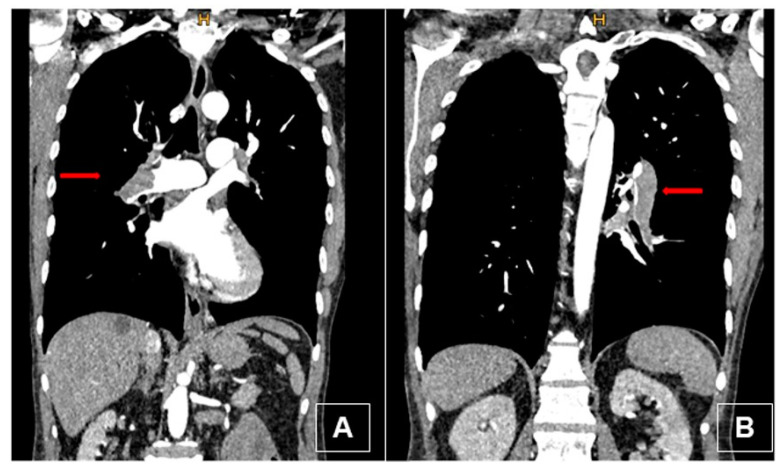
Preoperative computed tomography (coronal view) with intravenous contrast. (**A**) Thrombus in the right main pulmonary artery (red arrow) with typical signs of chronic disease: thrombus adherent to the wall, bronchial artery collaterals, webs. (**B**) Thrombus in the left lower lobe pulmonary artery (red arrow).

**Figure 4 jcm-11-04224-f004:**
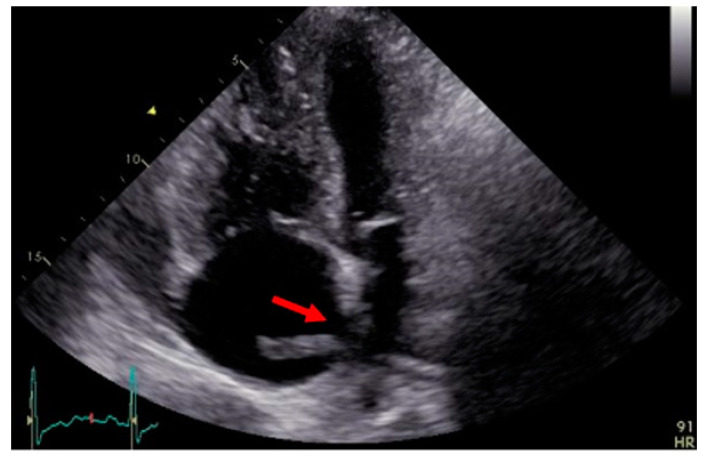
Transthoracic echocardiography showing the thrombus through the foramen ovale (red arrow).

**Figure 5 jcm-11-04224-f005:**
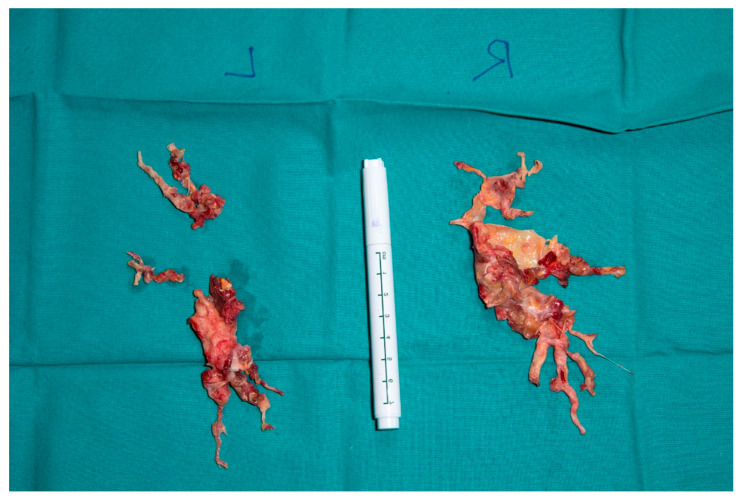
Right and left sided thrombendartherectomy specimen. (**Right**): Rudimentary upper lobe specimen and lower lobe branches are depicted, and fresh emboli chronic material is clearly visible (red arrows). (**Left**): Superior segment and basal lower lobe segment specimen.

**Figure 6 jcm-11-04224-f006:**
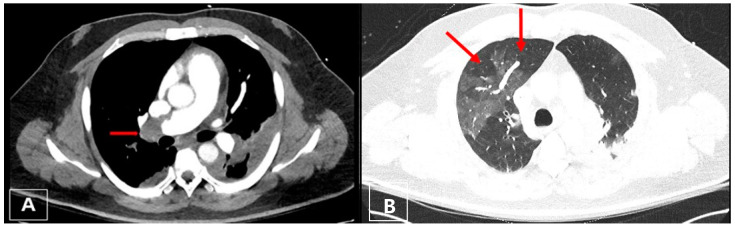
(**A**) Axial CT image with a central thrombus in the right pulmonary artery (red arrow); (**B**) right lung showing mosaic pattern whereas lucent areas correspond to perfusion defect (right upper lobe, red arrow).

**Figure 7 jcm-11-04224-f007:**
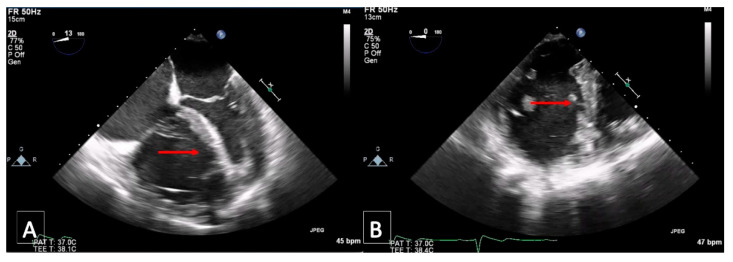
Transesophageal echocardiography: D-shaping (red arrow) in midesophageal 4 chamber (**A**,**B**); transgastric short-axis views at end systole (**A**) and end diastole (**B**).

**Figure 8 jcm-11-04224-f008:**
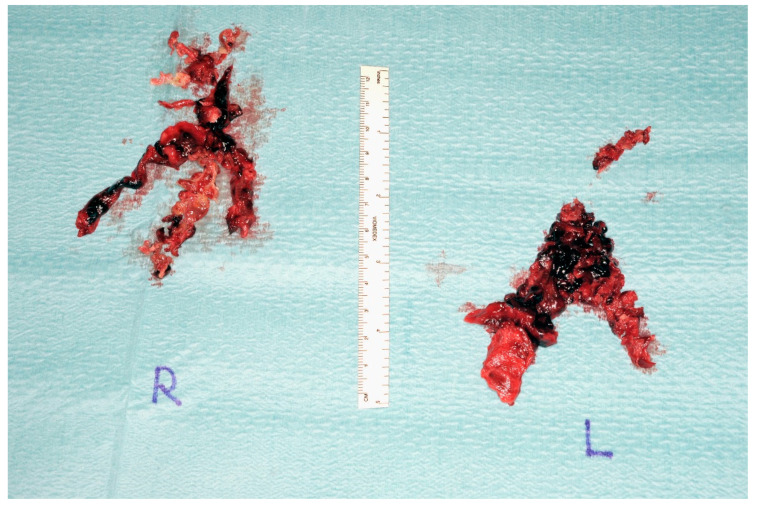
Postoperative endarterectomy specimen from both sides (**right**) and (**left**).

**Figure 9 jcm-11-04224-f009:**
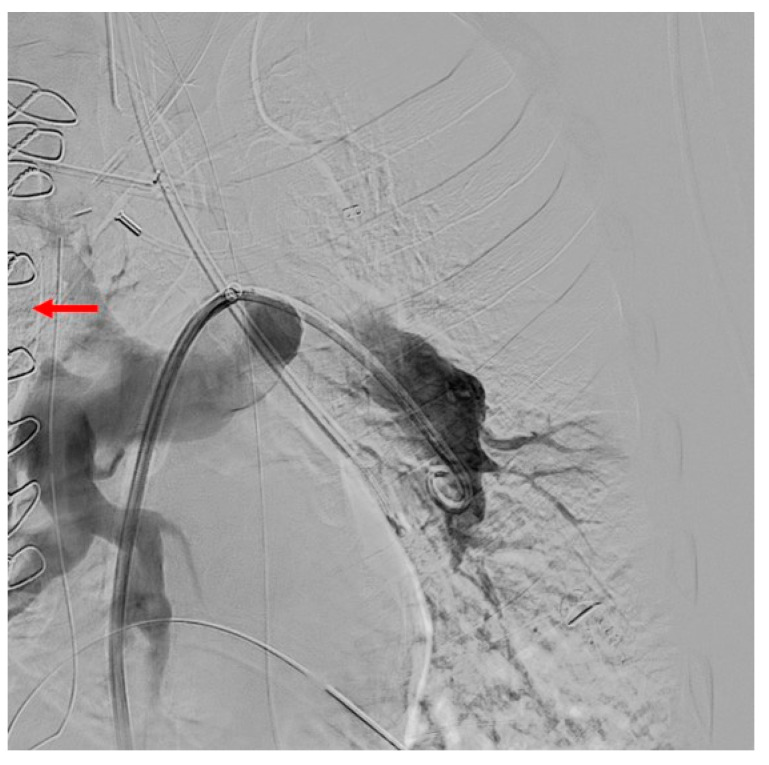
Persistent lack of contrast filling of the left upper lobe artery as well as presenting of the central re-thrombosis (red arrow) after surgery (PEA).

**Figure 10 jcm-11-04224-f010:**
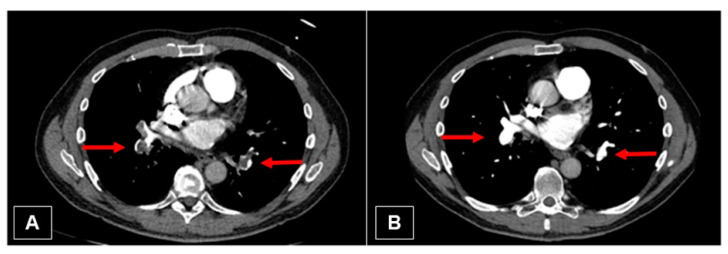
(**A**) Initial CT image with embolism of the right and left lower pulmonary artery (red arrows). (**B**) Nearly complete resolving of the thrombus (red arrows).

**Figure 11 jcm-11-04224-f011:**
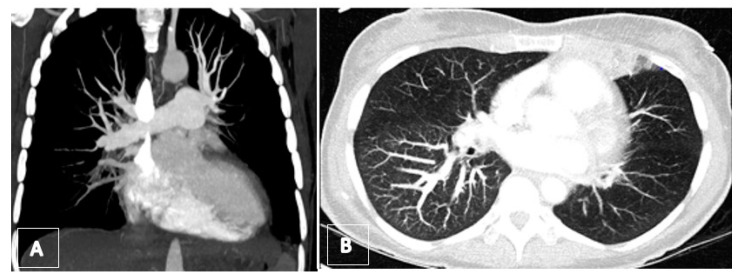
(**A**) Preoperative computed tomography angiography (coronar view) showing total occlusion of lower lobe arteries. (**B**) Axial maximum intensity projection (MIP) revealing the mosaic pattern with rarefication of vessels on left side as well as a hemorrhagic infarct in the lingula.

**Figure 12 jcm-11-04224-f012:**
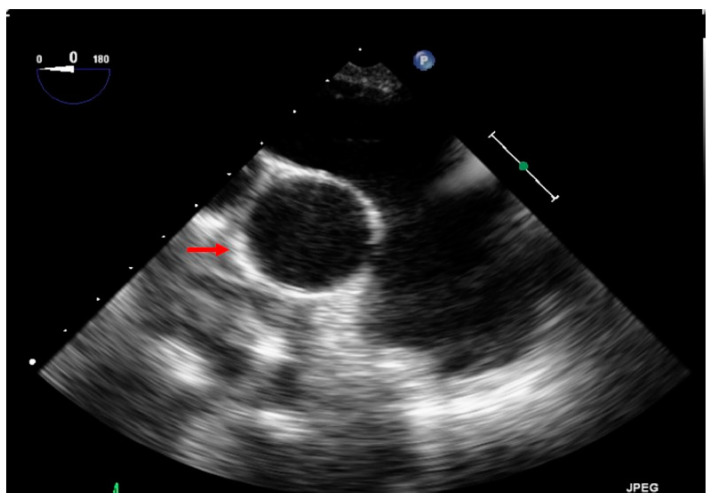
This echocardiography shows the dilated pulmonary artery in comparison to the aorta (red arrow on the aorta).

**Figure 13 jcm-11-04224-f013:**
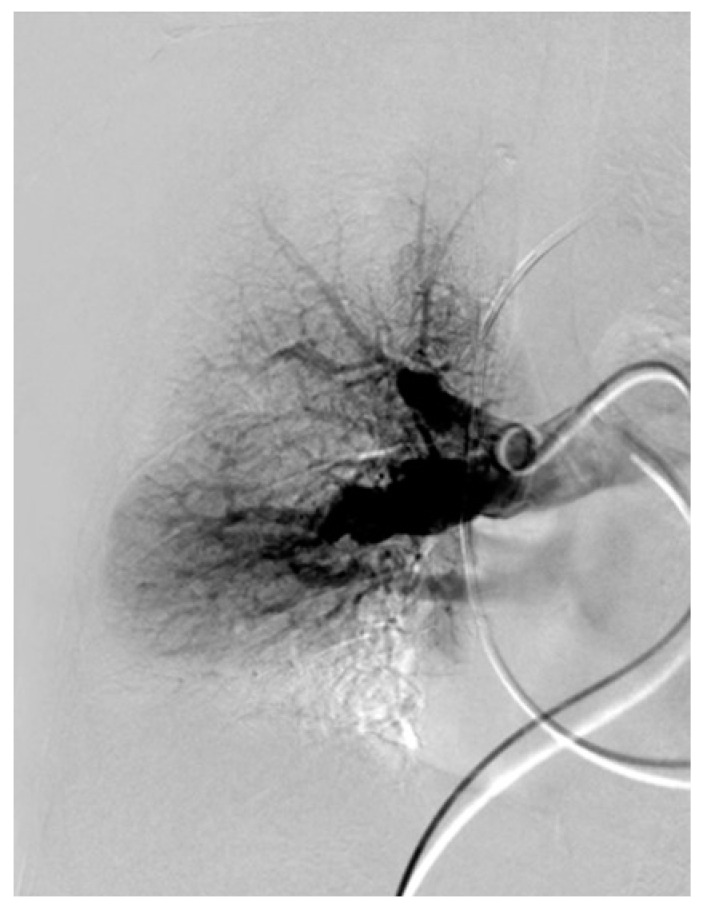
Angiography of the right pulmonary artery with webs of the apico-posterior artery of the upper lobe as well as peripheral.

**Figure 14 jcm-11-04224-f014:**
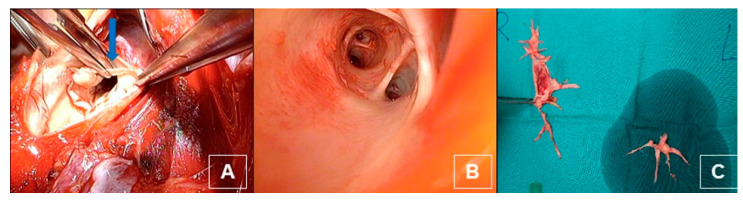
(**A**) Intraoperative dissection of the intima and media of the pulmonary artery and the branches (blue arrow). (**B**) After dissection with view on the middle and lower lobe artery. (**C**) Postoperative thrombendarterectomy sepcimen from both sides.

**Figure 15 jcm-11-04224-f015:**
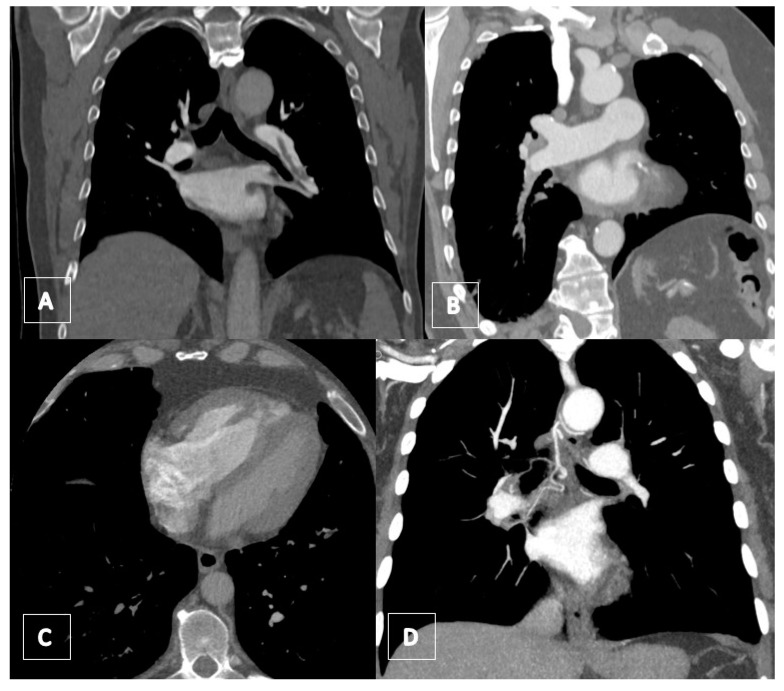
(**A**) Coronal CT image showing the free-floating embolus in the dilated pulmonary artery of the lower lobe surrounded by contrast agent. (**B**) Coronal CT image showing a chronic obstruction of the pulmonary artery of the right lower lobe with stricture of the vessel. (**C**) Axial CT image with contrast showing a right heart hypertrophy with thickened myocardium (>4 mm). RV/LV ratio is >1. (**D**) Coronal CT image demonstrating a dilated bronchial artery.

**Table 1 jcm-11-04224-t001:** Characteristics of surgical patients.

	Patient 1	Patient 2	Patient 4
**Age**	49 years	48 years	52 years
**Gender**	male	male	female
**Comorbidities/associated events**	Atrial septal defect	-	Recurrent hemoptysis, anticoagulation therapy for known CTEPH
**Acute event-surgery interval**	2 days	14 days	53 days
**HR at acute event**	90 bpm	89 bpm	75 bpm
**sPAP at acute event**	50 mmHg	54 mmHg	55 mmHg
**Medical treatment before surgery**	Double antiplatelet therapy, i.v. heparin	i.v: heparin, levosimendan, sildenafil, iloprost inhalation, antiplatelet	Sildenafil, macitentan, phenprocoumon
**Extracorporeal membrane oxygenation**	ECMO	ECMO	No
**Post-operative length of stay**	36 days	29 days	11 days
**Complications**	SIRS, septal hematoma, atrial fibrillation and flutter, atelectasis, delirium	sepis, aspergillosis, pneumonia, recurrent hematothorax, acute kidney failure, reperfusion edema, recurrent atelectasis, wound infection, groin abscess, femoral artery dissection, fingers and toes necrosis, SIRS and multi organ failure, death	delirium
**HR at 1-year follow-up**	78 bpm	-	66 bpm
**sPAP at 1-year follow-up**	28 mmHg	-	40 mmHg
**mPAP/PVR at 1-year**	18 mmHg/1.3 WU		27 mmHg/3.2 WU
**NYHA class at 1-year follow-up**	I	-	I

HR: heart rate; sPAP: systolic pulmonary artery pressure; NYHA: New York Heart Association; SIRS: systemic inflammatory response syndrome; ECMO: extracorporeal membrane oxygenation; CTEPH: chronic thromboembolic pulmonary hypertension.

## Data Availability

Not applicable.

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
