# Peer review of "Acute on Chronic Thromboembolic Pulmonary Hypertension: Case Series and Review of Management"

_jcm, 2022, doi:10.3390/jcm11144224_

Round 1
Reviewer 1 Report
In the present work, the authors address a very important topic of acute on chronic thromboembolic pulmonary hypertension. Although the incidence is relatively low, this pathology is underdiagnosed in most cases, especially in non-specialized centers. The authors describe 4 challenging clinical cases and review the literature, highlighting misdiagnosis and overlapping, specific findings on imaging, CTPA, transthoracic echocardiography. The manuscript is well written and the clinical cases presented here are very interesting and challenging even for highly specialized PERT teams. Although minor, I have a few points for the authors to consider:
1. page 2 - treatment of acute PE - the authors describe anticoagulation and thrombolytic rescue therapy. Please also include surgery (acute pulmonary thrombectomy) as a treatment option.
2. the specimen removed in case 1 (Figure 5) does not appear to me to be chronic (typical scar tissue), but a few weeks old organized thrombus followed by a fresh repeated embolism. Also, the need for postoperative ECMO suggests acute perioperative decompensation of the "untrained" right ventricle to elevated pulmonary pressure. Please comment on that.
3. In case 2 the authors opted for lysis therapy for re-thrombosis of the left pulmonary artery. In general, I think lysis therapy is somewhat risky in fresh operated patients with ongoing VVA-ECMO therapy. Why did the surgeons not opt for re-embolectomy? Please justify your decision and discuss it in section 3.7 "Case discussion: decision-making process".
4. Case 4 can also be interpreted as a subacute pulmonary embolism for which surgery was decided due to worsening symptoms. The specimen removed clearly shows chronic lesions, and the treatment strategy chosen in this particular case is fully justified. However, I would encourage the authors to discuss what would be the best timing for surgery in the case of subacute pulmonary embolism, with respect to the reorganization of the thrombus material and fragility of the thrombus.
Author Response
Question 1: page 2 - treatment of acute PE - the authors describe anticoagulation and thrombolytic rescue therapy. Please also include surgery (acute pulmonary thrombectomy) as a treatment option.
Answer:1: Thank you for this remark, we added the surgical option in the paragraph as mentioned by the current ESC guidelines
Changes 1: surgical option added in the paragraph
Question 2: the specimen removed in case 1 (Figure 5) does not appear to me to be chronic (typical scar tissue), but a few weeks old organized thrombus followed by a fresh repeated embolism. Also, the need for postoperative ECMO suggests acute perioperative decompensation of the "untrained" right ventricle to elevated pulmonary pressure. Please comment on that.
Answer: Thank you for this careful reading and highlighting this issue. However, there are macroscopic components that may indicate a chronic process, as described in the legend of figure 5.
We assumed that the stress of the acute event plus the surgery were the reason for deompensated right ventricle.
Question 3: In case 2 the authors opted for lysis therapy for re-thrombosis of the left pulmonary artery. In general, I think lysis therapy is somewhat risky in fresh operated patients with ongoing VVA-ECMO therapy. Why did the surgeons not opt for re-embolectomy? Please justify your decision and discuss it in section 3.7 "Case discussion: decision-making process".
Answer: Thank you. Due to the rather marginal general condition and the excessive burden of surgery, the decision was made in favor of lysis therapy, although this is also associated with an increased risk profile. We added this sentence in line 417-420.
- Case 4 can also be interpreted as a subacute pulmonary embolism for which surgery was decided due to worsening symptoms. The specimen removed clearly shows chronic lesions, and the treatment strategy chosen in this particular case is fully justified. However, I would encourage the authors to discuss what would be the best timing for surgery in the case of subacute pulmonary embolism, with respect to the reorganization of the thrombus material and fragility of the thrombus.
- Answer: 2-12 weeks of onset of symptoms is defined as subacute pulmonary embolism. Within this period, cardiovascular collapse and pulmonary angiogram showing massive pulmonary embolism (>50% obstruction in the major pulmonary arteries) are still missing. Although, surgical intervention is usually planned in a non-acute, recompensated phase, there is no optimal time for a surgical intervention in case of subacute embolism, and strongly depends on the patient´s general condition and already cardiovascular sequelae. Therefore, at this stage, drug therapy is the frontrunner. In exceptional cases, which are characterized by a rapid clinical deterioration, surgery is preferable despite the risk of reorganization of the thrombus material and the fragility of the thrombus.
We added this paragraph in line 436-444.
Reviewer 2 Report
This is a very well written case series and review of management. Authors clarity of thoughts on this topic is commendable. This study highlights important point about misdiagnosis or late diagnosis of CTEPH which as a clinician we all see in our practice. Good images with nice explanation in figure legends. I have few minor questions:
In Case 1: A) was there a Right heart Cath done before the patient was taken to operating room to confirm PA pressures? If yes, what were the pressures? B) There is mention that CT scan showed findings of underlying CTEPH. Can authors elaborate on these findings for the readers. C) Was patient discharged on any anticoagulation medication? If yes, which one and what was the planned duration?
Case 3: Authors mentioned that patient lacked signs of PH. This part is bit unclear to me. What are they referring too? What signs were lacking? B) Was a right heart Cath done for this patient along with echocardiogram during the initial presentation?
Case 4: Nice save in this challenging case of diffuse alveolar hemorrhage and CTEPH. It would be interesting to know did team decide to send her on anticoagulation after the surgery? It can be challenging to do that when she presented with hemoptysis. But, would love to know what authors decided to do.
Author Response
Case 1:
Question A) was there a Right heart Cath done before the patient was taken to operating room to confirm PA pressures? If yes, what were the pressures?
Answer A. We did not perform a right heart catheterization before the operation since there was evidence of elevated PA pressure already at the TTE (50 mmHg). This finding, along with the presence of the thrombus into the PFO and the severe clinical condition of the patient, were considered as sufficient to indicate a surgical intervention without further evaluation.
Changes A: explanation added in case 1 description
Question B) There is mention that CT scan showed findings of underlying CTEPH. Can authors elaborate on these findings for the readers
Answer B: presence of wall-adjacent eccentric filling defects of the pulmonary vessels, enlargement of the pulmonary artery and hypertrophy of the right ventricle with D-shape.
Changes B: description added in case 1
Question C) Was patient discharged on any anticoagulation medication? If yes, which one and what was the planned duration?
Answer C. The patient was discharged on Marcoumar (Phenprocoumon) which was switched to rivaroxaban sine die after 5 years
Changes C: information added
Case 3:
Question A: Authors mentioned that patient lacked signs of PH. This part is bit unclear to me. What are they referring too? What signs were lacking?
Answer A: we refer to symptoms and clinical signs typical of established PH
Changes A: specification added
Question B) Was a right heart Cath done for this patient along with echocardiogram during the initial presentation?
Answer B: the right earth cath was not performed at presentation. He has been followed-up with TTE and the first invasive assessment was done done after 5 months of oral anticoagulation therapy and repeated after 1 year from the acute event.
Changes B: information added in the text